# The prevalence of hepatitis C virus in hemodialysis patients in Pakistan: A systematic review and meta-analysis

Sohail Akhtar[1]*, Jamal Abdul Nasir[1], Muhammad Usman[2,3], Aqsa Sarwar[1], Rizwana Majeed[1], Baki Billah[4]

**1** Department of Statistics, Government College University Lahore, Lahore, Pakistan, **2** Department of Statistics, University of Peshawar, Peshawar, Pakistan, **3** Diabetes Research Centre, University of Leicester, Leicester United Kingdom, **4** School of Public Health and Preventive Medicine, Monash University, Melbourne, Australia

* s.akhtar@gcu.edu.pk, akhtar013@gmail.com

## Abstract

**Data Availability Statement:** All relevant data are within the paper and its Supporting Information files.

### Background

Hepatitis C virus (HCV) infection is one of the most common bloodborne viral infections reported in Pakistan. Frequent dialysis treatment of hemodialysis patients exposes them to a high risk of HCV infection. The main purpose of this paper is to quantify the prevalence of HCV in hemodialysis patients through a systematic review and meta-analysis.

### Methods

We systematically searched PubMed, Medline, EMBASE, Pakistani Journals Online and Web of Science to identify studies published between 1 January 1995 and 30 October 2019, reporting on the prevalence of HCV infection in hemodialysis patients. Meta-analysis was performed using a random-effects model to obtain pooled estimates. A funnel plot was used in conjunction with Egger's regression test for asymmetry and to assess publication bias. Meta-regression and subgroup analyses were used to identify potential sources of heterogeneity among the included studies. This review was registered on PROSPERO (registration number CRD42019159345).

### Results

Out of 248 potential studies, 19 studies involving 3446 hemodialysis patients were included in the meta-analysis. The pooled prevalence of HCV in hemodialysis patients in Pakistan was 32.33% (95% CI: 25.73–39.30; $I^2$ = 94.3%, $p < 0.01$). The subgroup analysis showed that the prevalence of HCV among hemodialysis patients in Punjab was significantly higher (37.52%; 95% CI: 26.66–49.03; $I^2$ = 94.5, $p < 0.01$) than 34.42% (95% CI: 14.95–57.05; $I^2$ = 91.3%, $p < 0.01$) in Baluchistan, 27.11% (95% CI: 15.81–40.12; $I^2$ = 94.5, $p < 0.01$) in Sindh and 22.61% (95% CI: 17.45–28.2; $I^2$ = 78.6, $p < 0.0117$) in Khyber Pukhtoonkhuwa.

**Funding:** The author(s) received no specific funding for this work.

**Competing interests:** The authors have declared that no competing interests exist.

## Conclusions

In this study, we found a high prevalence (32.33%) of HCV infection in hemodialysis patients in Pakistan. Clinically, hemodialysis patients require more attention and resources than the general population. Preventive interventions are urgently needed to decrease the high risk of HCV infection in hemodialysis patients in Pakistan.

## Introduction

Hepatitis C virus (HCV) infection is one of the most commonly reported viral infections in both developing and developed countries, causing significant mortality and morbidity and costing billions of dollars annually [1, 2]. The prevalence rate of HCV infection in hemodialysis varies substantially among different geographical regions [3–5]. Recent studies have shown that the HCV prevalence in hemodialysis patients varies from 1.4%–28.3% in developed countries and 4.7%–41.9% in developing countries [6]. Patients on hemodialysis are at a very high risk of HCV infection due to repeated blood transfusions, frequent hospitalization and infected hemodialysis units with HCV. HCV and its associated complications have a significant impact on the life expectancy of hemodialysis patients. Hemodialysis patients with HCV infection are at a higher risk of death than uninfected hemodialysis patients [7, 8].

Pakistan is a developing country, and, according to the human development index of the United Nations, it stands at 150th position out of 189 countries and territories. In the South Asian region, Pakistan's neighbours have a much lower human development index: Iran (60th), India (130th) and Bangladesh (136th) [9]. The health system in Pakistan is below international standards. Transfusion with HCV contaminated blood and dialysis units are the major risk factors for the spread of hepatitis C in hemodialysis patients. It is estimated that nearly 40% of blood transfusions in Pakistan are not screened for any infectious diseases [10].

Multiple studies have reported the prevalence of HCV infection among hemodialysis patients in Pakistan [11–29]. To the best of our knowledge, no official nationwide survey or national health registry has to date estimated the prevalence of HCV in hemodialysis patients in Pakistan. The prevalence of HCV among hemodialysis patients varies significantly among these published studies (from 16.8% to 68%) [12, 14]. This study aims to draw on the available published papers from Pakistan to systematically identify, select, review, summarize and estimate the pooled prevalence of HCV in hemodialysis patients. This study may aid in measuring the countrywide pooled prevalence of HCV in the absence of a national registry in Pakistan for the measurement of the prevalence of HCV among hemodialysis patients. The findings of this study may also aid in developing a management policy to reduce this perceived prevalence. This is the first systematic review and meta-analysis that estimate the pooled prevalence of HCV infection in hemodialysis patients in Pakistan.

## Methods

### Design

This study was performed using the Preferred Reporting Items for Systematic Reviews and Meta-Analyses (PRISMA) guidelines [30]. The protocol of this study was registered with the International Prospective Register of Systematic Reviews (PROSPERO), with registration number CRD42019159345.

## Search strategy

In this review, two authors (AS and RM) independently searched PubMed, Medline, EMBASE, Pakistani Journals Online and Web of Science to identify all articles published from 1 January 1995 to 30 October 2019, reporting on the prevalence of HCV infection in hemodialysis patients in Pakistan. We searched using keywords such as 'HCV', 'Hepatitis C', 'dialysis', 'hemodialysis', 'prevalence' and 'Pakistan'; variations of these terms were also searched. In addition, we searched the reference lists of the included articles to identify additional studies that were not detected by the electronic searches.

## Inclusion and exclusion criteria

The following inclusion and exclusion criteria were used in this study. Studies were included in the meta-analyses if they (1) were published in peer-reviewed journals only, (2) were conducted in Pakistan, (3) reported on the prevalence of HCV in hemodialysis patients, (4) were published in the English language and (5) focused on hemodialysis patients over the age of 18 years. Studies were excluded if they (1) were published in a non-English language, (2) were case series, reviews, letters and editorials or commentaries, (3) did not contain data on the prevalence of HCV in hemodialysis patients, (4) contained duplicate (overlapping) data (i.e. were used in more than one article; in such cases, the up-to-date data were considered) and (5) included Pakistani communities living outside Pakistan.

## Data collection

Two authors (AS and RM) independently extracted the data from the included studies onto a predefined data extraction form. The extracted information contained the following information: surname of the first author, year of publication, baseline study year, study geographical region, proportion of men, average age of hemodialysis patients, sampling design, sample size and methodological quality of each study. The authors agreed that they would settle their disagreement, if any, through discussion or referral to a third author (JAN).

## Methodological quality of the included studies

The methodological quality of the included studies was assessed through the tool developed by the Joanna Briggs Institute (JBI) [31]. The JBI tool consists of nine questions (see Appendix-1 for details). For each question, a score was assigned (0 for 'yes' and 1 for 'no'); the scores were summarized across the items to attain a total quality score that ranged from 0 to 9. Studies were then categorized according to the awarded points; a point of 7–9, 5–7 or 0–4 was rated as having a high, medium or low risk of bias, respectively. Two authors (AS and RM) independently assessed the methodological quality of each included study. They agreed to settle their disagreement, if any, by mutual consensus or referral to a third author (JAN) for a final decision. The checklist for the methodological quality appraisal of the included studies is presented in the supplementary file (S1 Appendix).

## Statistical analyses

Meta-analysis was conducted using statistical software R, version 3.5.2 [32]. We used the 'meta' and 'metafor' packages in R to pool the prevalence across the studies, which was performed using random-effects models of the DerSimonian and Laird method. A forest plot was used to visually display the prevalence estimates with their corresponding 95% confidence intervals (CIs). In the presence of heterogeneity (as expected and observed), random-effect models have better properties and are more conservative than fixed-effect models [33,34]. The

Freeman–Tukey double arcsine transformation was used to stabilize the variance of the raw prevalence of each included study [35]. Heterogeneity among the included studies was evaluated using the Cochran Q test and quantified using $I^2$ statistic [36, 37]. The heterogeneity among the studies was categorized as $I^2$-values of 75%, 50% and 25%, which were considered as having high, moderate and low levels of heterogeneity, respectively [38, 39]. Statistical significance was considered at a $p$-value of less than 0.10 using 2-tailed tests. To explore the possible reasons for heterogeneity, subgroup analyses and meta-regression were conducted by geographical region, sample size, year of publication, year of data collection, gender and average age of the patients. A funnel plot and Egger's regression test were used to investigate the presence of publication bias [40], with a $p$-value of < 0.10 being considered as statistically significant. We also used the 'Trim and Fill' procedure (nonparametric method) to further evaluate the asymmetry of the funnel plot [38].

## Results

### Literature search

We initially identified 248 potential articles from a comprehensive literature search. After the elimination of duplicates, 73 articles remained. We screened the titles and abstracts and excluded 31 irrelevant articles. We scrutinized the full text of the remaining 42 articles for eligibility, of which 23 were excluded with valid reasons. Finally, only 19 articles fulfilled the inclusion criteria, whose data were extracted accordingly. Drawing on the PRISMA flow diagram [30], the flow diagram of the study inclusion process is presented in Fig 1. The PRISMA checklist is presented in the supplementary file (S1 Checklist).

### Characteristics of the selected studies

The details and main characteristics of the 19 selected studies [11–29] are presented in Table 1. Twelve studies had used a cross-sectional research design, while seven studies did not explicitly specify their research design. Nine studies had used a convenient sampling strategy to select their representative sample while the other nine studies did not explicitly describe their sampling procedure; only one study had used a random sampling strategy. The number of hemodialysis patients per study ranged from 28 to 500, with a total of 3446 patients across all studies. The included articles were published between 2002 and 2019, while the period of participant inclusion was from 1999 to 2018. Four geographical regions (provinces) of Pakistan were represented in the articles: three studies were conducted in Sindh, 10 studies in Punjab, four studies in Khyber Pukhtoonkhuwa and two studies in Baluchistan. The average duration of dialysis of hemodialysis patients was reported in nine studies. Most of the studies (9 out of 19) reported the HCV prevalence using the results from the ELIZA (enzyme-linked immunosorbent assay) test. Only five studies reported the confirmation of HCV infection by the RNA (Ribonucleic acid) test. Two studies used the CILA (chemiluminescence immunoassay) method for the confirmation of HCV. Three studies did not explicitly refer to any type of test used for the HCV antibody. The proportion of male participants ranged from 14.38% to 71.13%. The average age of participants ranged from 36.5 to 55.2 years. Thirteen articles had reported the gender of their participants. After assessing the methodological quality of the studies, 15 were found to have a low risk of bias, four had a medium quality, and no article was found with poor quality.

### Meta-analysis

All statistical analyses of the prevalence of HCV in hemodialysis patients are presented in Table 2. The pooled prevalence of HCV in hemodialysis was 32.33% (95% CI: 25.73–39.30) $I^2$

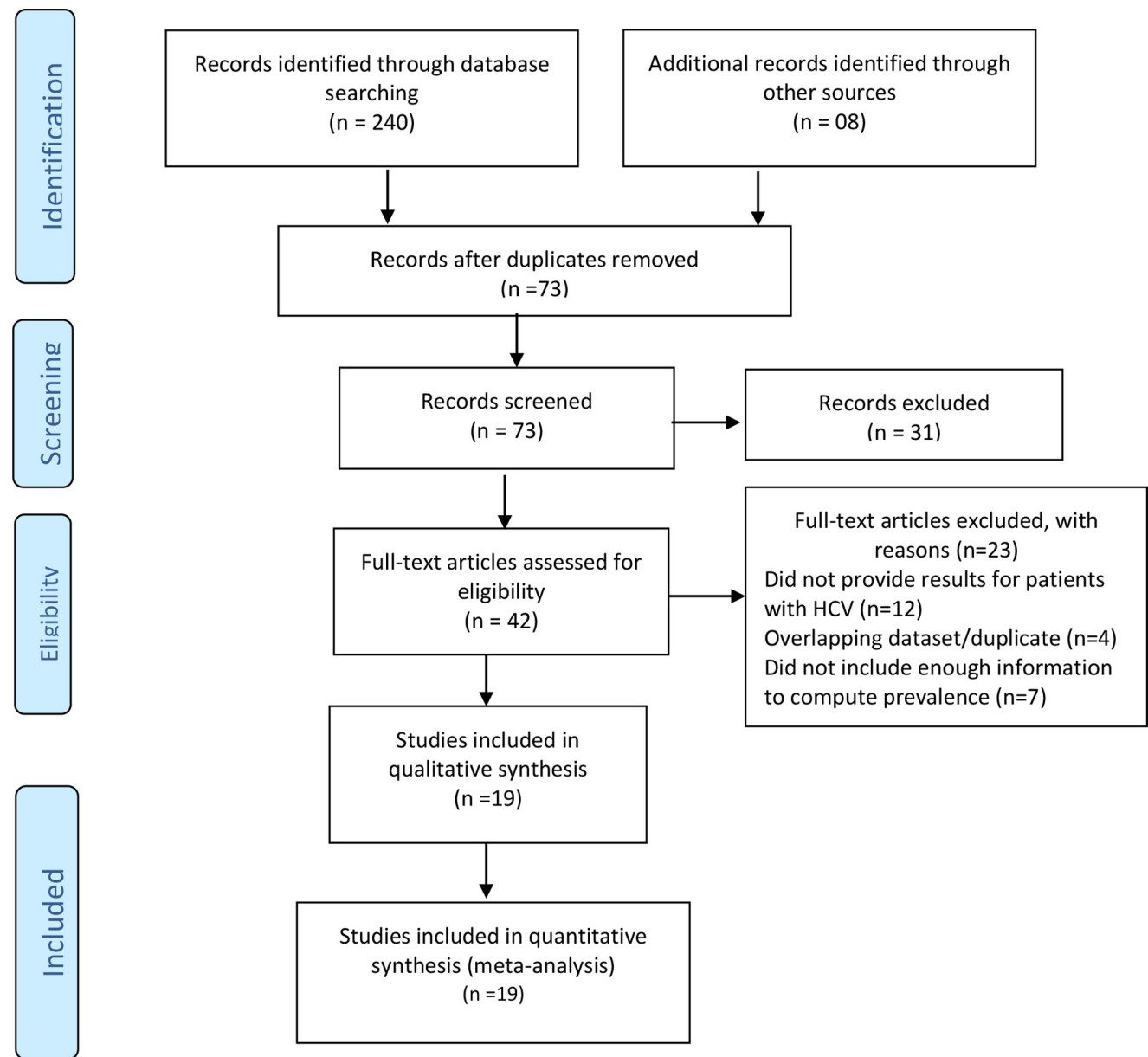

**Fig 1. Flow diagram of identification and selection of studies for inclusion in the meta-analysis, following the PRISMA 2009 guidelines [30].**

= 94.5%, based on 19 studies in a total sample of 3446 individuals. The graphical presentation of the pooled prevalence of HCV in hemodialysis patients is presented in the forest plot (Fig 2). The funnel plot (Fig 3) revealed no publication bias, which was confirmed by Egger's regression test ($p = 0.3154$). Furthermore, no publication bias in the analysis was confirmed by Trim and Fill sensitivity analysis, as we did not find any missing study.

## Heterogeneity and subgroup analysis

The subgroup analysis of the prevalence of HCV in hemodialysis patients is presented in Table 2. Initially, the analysis was stratified by gender, and it was found that it was not

**Table 1. Description and list of characteristics of the included studies.**

| Author | Year | Year of Data Collection | Province | Sampling Method | Study design | Method use to diagnose HCV | Sample size | Total infected people of HCV | Prevalence of HCV | % of male participant | Age (year) | Mean Duration of dialysis (months) | Methodological Quality |
|---|---|---|---|---|---|---|---|---|---|---|---|---|---|
| Butt et al. [11] | 2019 | 2017–2018 | Sindh | Convenient Sampling | NA | RNA | 80 | 31 | 38.750 | 35.48 | 36.5 | 40.44 | Low Risk Bias |
| Mahmud et al.[12] | 2014 | 2012–2013 | Sindh | Convenient Sampling | Cross-Sectional | CLIA | 189 | 31 | 16.402 | 49.7 | 51.88 | NA | Low Risk Bias |
| Chishti et al. [13] | 2015 | 2010–2011 | Sindh | Convenient Sampling | Cross-Sectional | ELIZA | 200 | 58 | 29.000 | 34.5 | NA | NA | Low Risk Bias |
| Gul et al. [14] | 2003 | 1999 | Punjab | Convenient Sampling | Cross-Sectional | NA | 50 | 34 | 68.000 | NA | NA | NA | Medium Risk Bias |
| Mumtaz et al. [15] | 2009 | 2008 | Punjab | NA | Cross-Sectional | NA | 50 | 14 | 28.000 | NA | 42.3 | NA | Medium Risk Bias |
| Anwar et al. 16] | 2016 | 2012–2013 | Punjab | Random Sampling | Cross-Sectional | RNA | 60 | 14 | 23.333 | 71.7 | NA | NA | Low Risk Bias |
| Khokhar et al. [17] | 2005 | 2002–2003 | Punjab | Convenient Sampling | Cross-Sectional | ELIZA | 97 | 23 | 23.711 | 66 | 54.26 | 34.8 | Low Risk Bias |
| Shafi et al. [18] | 2003 | 2000–2002 | Punjab | NA | NA | ELIZA | 122 | 24 | 19.672 |  | NA | NA | Low Risk Bias |
| Shafi et al. [19] | 2017 | NA | Punjab | Convenient Sampling | Cross-Sectional | ELIZA | 180 | 49 | 27.222 | 68.45 | 48.7 | 98.4 | Low Risk Bias |
| Shafi et al. [20] | 2002 | 2001–2002 | Punjab | NA | NA | ELIZA | 190 | 47 | 24.737 | 36.32 | 38.6 | 29.3 | Low Risk Bias |
| Ismail et al.[21] | 2016 | 2016–2016 | Punjab | Random Sampling | Cross-Sectional | NA | 190 | 93 | 48.947 | 70 | 43.68 | 25.46 | Medium Risk Bias |
| Kiani et al. [22] | 2018 | 2016 | Punjab | Convenient Sampling | Cross-Sectional | ELIZA | 201 | 128 | 63.682 | Na | NA | 4.5 | Low Risk Bias |
| Hussain et al. [23] | 2019 | 2016–2017 | Punjab | NA | NA | ELIZA | 230 | 123 | 53.478 | 30 | 49.7 | NA | Low Risk Bias |
| Ali et al. [24] | 2011 | NA | Khyber Pakhtunkhwa | NA | NA | RNA | 28 | 7 | 25.000 | NA | NA | NA | Low Risk Bias |
| Khan et al. [25] | 2011 | 2010 | Khyber Pakhtunkhwa | Convenient Sampling | NA | RNA | 384 | 112 | 29.167 | 63.557 | 40.9 | 80.4 | Low Risk Bias |
| Ali et al. [26] | 2019 | 2013–2014. | Khyber Pakhtunkhwa | Convenient Sampling | Cross-Sectional | RNA | 480 | 94 | 19.583 | 14.38 |  | NA | Low Risk Bias |
| Anjum et al. [27] | 2015 | 2014–2015 | Khyber Pakhtunkhwa | NA | Cross-Sectional | ELIZA | 500 | 98 | 19.600 | 68.1 | 46 | NA | Low Risk Bias |
| Zarkoon et al. [28] | 2008 | 2006–2007 | Baluchistan | Convenient Sampling | Cross-Sectional | ELIZA | 97 | 23 | 23.711 | 71.132 | 55.2 | 34.8 | Medium Risk Bias |
| Lodi et al. [29] | 2019 | 2018 | Baluchistan | NA | Cross-Sectional | CLIA | 118 | 54 | 45.76 | 60.1 | 43.02 | NA | Low Risk Bias |

**Table 2. Prevalence of HCV among Hemodialysis patients in Pakistan, from January 1995 to Octuber 2019.**

| Characteristics | Studies | Sample | Cases | Prevalence, % (95%CI) | $I^2$, % | Heterogeneity | P-Egger test | P-Difference |
|---|---|---|---|---|---|---|---|---|
| **Prevalence of HCV in Hemodialysis patients** | 19 | 3446 | 1057 | 32.33 (25.73–39.2) | 94.3 | < 0.001 | 0.4417 | |
| **Time Period** | | | | | | | | 0.2063 |
| 2002–2008 | 5 | 556 | 151 | 30.43 (18.68–43.61) | 90.1 | < 0.001 | | |
| 2009–2016 | 4 | 651 | 164 | 24.04 (16.37–32.62) | 75.0 | < 0.001 | | |
| 2017–2019 | 10 | 2239 | 742 | 36.37 (26.00–47.40) | 96.3 | < 0.001 | | |
| **Gender** | | | | | | | 0.9696 | 0.9818 |
| Male | 6 | 540 | 174 | 33.92 (20.32–48.96) | 78.6 | < 0.001 | | |
| Female | 6 | 290 | 114 | 33.85 (24.04–44.36) | 59.7 | < 0.001 | | |
| **By Province** | | | | | | | 0.4417 | 0.0946 |
| Punjab | 10 | 1253 | 503 | 37.51 (26.66–49.03) | 94.5 | < 0.001 | | |
| Baluchistan | 2 | 550 | 112 | 34.42 (14.95–57.05) | 91.3 | < 0.001 | | |
| Sindh | 3 | 850 | 236 | 27.11 (15.81–40.12) | 88.3 | < 0.001 | | |
| Khyber Pakhtunkhwa | 4 | 793 | 206 | 22.61 (17.44–28.22) | 78.6 | < 0.001 | | |
| **By dignoistic method** | | | | | | | 0.4417 | 0.2059 |
| RNA | 5 | 1032 | 258 | 26.62 (19.81–34.01) | 78.6 | < 0.001 | | |
| CLIA | 2 | 307 | 85 | 29.91 (6.44–61.20) | 96.7 | < 0.001 | | |
| ELIZA | 9 | 1817 | 573 | 31.14 (21.02–42.24) | 95.8 | < 0.001 | | |
| NA (method not clear) | 3 | 290 | 141 | 48.24 (29.66–67.06) | 87.9 | < 0.001 | | |

statistically significant: the pooled prevalence of HCV in male hemodialysis patients was 33.92% (95% CI: 20.32–48.96, $I^2$ = 78.6%), and the pooled prevalence of female HCV in hemodialysis patients was 33.85% (95% CI: 24.04–44.36; $I^2$ = 78.6%). Across regions, a significant difference was observed between provinces: the pooled prevalence of HCV in hemodialysis patients was 37.51% (95% CI: 26.66–49.04) in Punjab, which was higher than the pooled prevalence in Baluchistan (34.42%; 95% CI: 14.95–57.05), in Sindh (27.11%; 95% CI: 15.81–40.12) and in Khyber Pakhtunkhwa (22.61%; 95% CI: 17.44–28.22). Furthermore, the pooled prevalence of HCV in hemodialysis patients was stratified by three publication periods of 2002–2008, 2009–2016 and 2017–2019. The prevalence of HCV among hemodialysis patients was 30.43% (95% CI: 18.68–43.61) in the first period, 24.03% (95% CI: 16.36–32.62) in the second period and 36.36% (95% CI: 26.00–47.41) in the third period. Lastly, the pooled prevalence of HCV in hemodialysis patients was stratified using the diagnostic methods of HCV: RNA (26.62%; 95% CI: 19.81–34.01), CILA (29.91%; 95% CI: 6.44–61.20), ELIZA (31.14%; 95% CI: 21.02–42.24) and the unstated method NA (48.24%; 95% CI: 29.66–67.06).

No publication bias was noticed in any subgroup analyses. The univariate meta-regression revealed that the pooled prevalence of HCV among hemodialysis patients was not associated with the year of publication, year of data collection, male proportion, mean age of hemodialysis patients, sample size and duration of dialysis.

## Discussion

The main objective of this systematic review and meta-analysis was to summarize all available published data on the prevalence of HCV in hemodialysis patients of Pakistan. The information provided in this study may play a positive role in improving public health interventions in the country, as there is no national registry to measure the prevalence of HCV in hemodialysis patients in Pakistan. Therefore, this study may help decrease the incidence of HCV in hemodialysis patients in Pakistan. Nineteen studies based on 3446 hemodialysis patients were

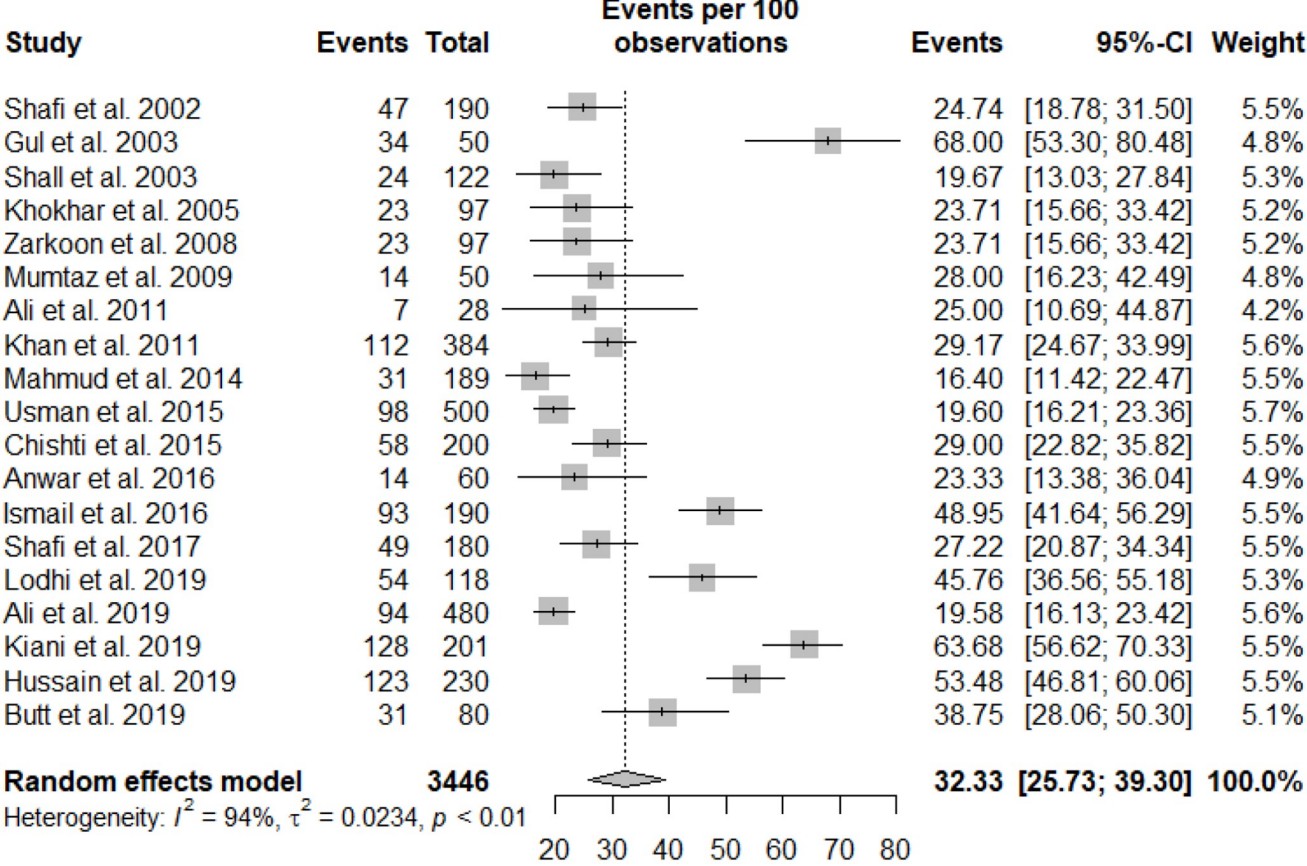

**Fig 2. Forest plot of prevalence of HCV in hemodialysis patients in Pakistan January 1995 to October 2018.**

included in this study. The pooled HCV prevalence among hemodialysis patients in Pakistan is 32.33%, which is five times higher than the prevalence of HCV in the general Pakistani population (6.2%) [41]. This means that every third hemodialysis patient is infected with HCV in Pakistan. This may be due to a lack of education and awareness of HCV transmission, a lack of scientifically and medically qualified personnel, a lack of proper health infrastructure (e.g. the use of unsterilized instruments), non-adherence or gaps in the implementation of practices recommended by the World Health Organization (WHO), inadequate use of erythropoietin or inadequate screening of HCV for donated blood [36, 42, 43]. The pooled prevalence of HCV in hemodialysis patients in Pakistan is almost three times higher than that of a similar study (meta-analysis) conducted in neighbouring Iran (11%) [44], nearly two times higher than Taiwan (17.3%) [45] and 18.8% in India [46].

The subgroup analysis revealed that HCV infection prevalence among the hemodialysis patients was observed across all provinces in Pakistan except Gilgit-Baltistan, as we did not find any studies for this province. Our results show that the prevalence of HCV among hemodialysis patients is higher in Punjab (37.51%) than in Sindh (27.11%), Baluchistan (23.71%) and Khyber Pakhtunkhwa (22.61%). This variability may be due to differences in ethnicity, health provision system and characteristics of the study population.

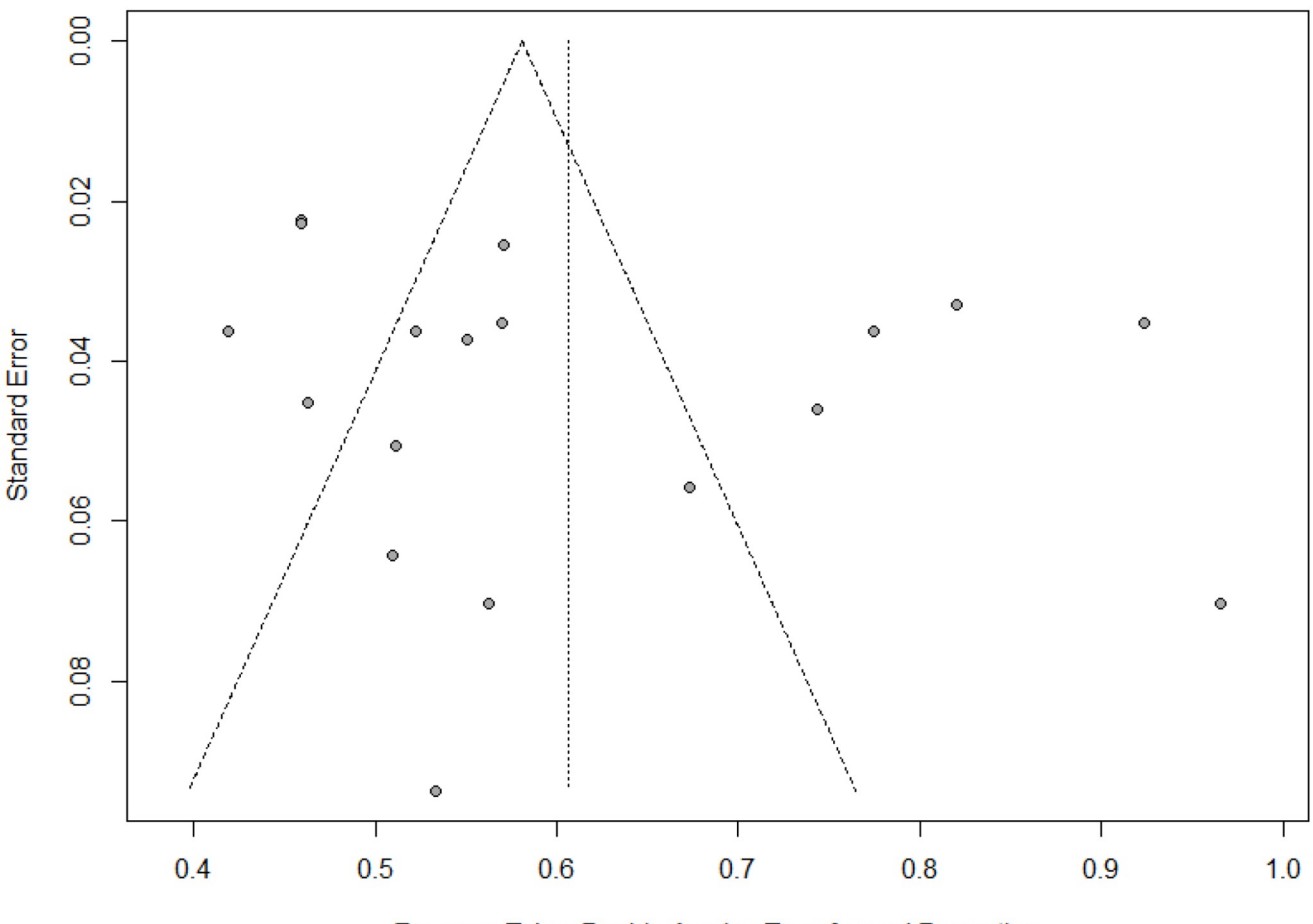

**Fig 3. Funnel plot of the prevalence HCV in hemodialysis patients in Pakistan January 1995 to October 2018.**

It was also observed that the prevalence of HCV does not appear to be decreasing with time in Pakistan (from 30.43% in 2002–2008 to 36.37% in 2015–2019). This is because, contrary to the worldwide trend, the prevalence of HCV in the general population of Pakistan is increasing gradually [41]. Also, in developing countries, proper techniques and infection control practices are often inadequate, and the quality of medical care is often poor [47].

Our results also demonstrated that the pooled prevalence of HCV hemodialysis patients is almost similar between males (33.92%) and females (33.85%). Furthermore, meta-regression analyses showed that the changes in the prevalence of HCV among hemodialysis patients over the past two decades have not been statistically significant (i.e. considering both year of publication and year of data collection). The average age of hemodialysis patients is insignificant compared with the prevalence of HCV. Rather than age, it is the number of dialysis patients that plays a vital role in the prevalence of HCV. Currently, we do not have any data on this variable.

To the best of our knowledge, this is the first systematic review and meta-analysis to summarize all available data on the prevalence of HCV infection in hemodialysis patients in

Pakistan. The strengths of this review are its use of a systematic and comprehensive literature search strategy with a double review process with the participation of two independent authors in the whole review process and data extraction. In addition, any disagreement between the two investigators about the extracted information was resolved by a third researcher to improve the quality of this analysis. No publication bias was found in our analysis, which suggests that we are unlikely to have missed any significant studies that could have influenced the results. Furthermore, the methodological quality of all the articles revealed a low-risk bias. As illustrated by the meta-regression analysis, the methodological quality of the studies had an insignificant effect on pooled prevalence estimates. Four major provinces of Pakistan were represented in the determination of HCV prevalence in hemodialysis patients.

This study has several limitations. First, most of the studies had a small sample size with a pooled sample size of 3446. Second, only univariate meta-regression analysis was used. We had intended to use a multivariable meta-regression analysis by considering all the factors simultaneously; however, it was not possible to use a multivariable meta-regression analysis due to the small number of studies. Third, our estimates showed significant heterogeneity, especially in the meta-analyses. This is likely that other causes of variability may have been missed in our analysis, such as the frequency of dialysis, other diseases and genetic factors, which we were not able to test due to data unavailability in the articles.

## Conclusion

The pooled prevalence of HCV infection among hemodialysis patients in Pakistan was 32.33%; however, this rate varies from province to province. The observed prevalence is higher than in neighbouring countries, such as Iran and Bangladesh. Pakistan is a developing country and lacking in resources for appropriate stylized dialysis units as well as facilities in dialysis centres and hospitals. Special health education programmes for both patients and healthcare staff are required, and standard screening tests should be carried out before dialysis is performed.

## Supporting information

**S1 Appendix. JBI critical appraisal checklist applied for included studies in the systematic review.**
(DOCX)

**S1 Checklist. PRISMA 2009 checklist (adapted for KIN 4400).**
(DOC)

## Author Contributions

**Conceptualization:** Sohail Akhtar.

**Data curation:** Jamal Abdul Nasir, Aqsa Sarwar, Rizwana Majeed.

**Formal analysis:** Sohail Akhtar, Aqsa Sarwar, Rizwana Majeed.

**Investigation:** Sohail Akhtar, Jamal Abdul Nasir, Aqsa Sarwar, Rizwana Majeed.

**Methodology:** Jamal Abdul Nasir, Muhammad Usman.

**Software:** Sohail Akhtar.

**Supervision:** Sohail Akhtar, Muhammad Usman.

**Visualization:** Muhammad Usman.

**Writing – original draft:** Sohail Akhtar, Jamal Abdul Nasir, Muhammad Usman.

**Writing – review & editing:** Baki Billah.

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
