## [Decision Letter · Decision Letter 0]

17 Feb 2020

PONE-D-20-00973

The prevalence of hepatitis C virus in hemodialysis patients in Pakistan: a systematic review and meta-analysis

PLOS ONE

Dear Dr. Akhtar,

Thank you for submitting your manuscript to PLOS ONE. After careful consideration, we feel that it has merit but does not fully meet PLOS ONE’s publication criteria as it currently stands. Therefore, we invite you to submit a revised version of the manuscript that addresses the points raised during the review process.

In addition to the comments raised by reviewers, please discuss the prevalence of HCV infection among HD patients between Pakistan and the other countries in Asian-Pacific regions, especially Taiwan where incidence and prevalence of uremia ranking in the top 3 countries in the world.

We would appreciate receiving your revised manuscript by Apr 02 2020 11:59PM. To enhance the reproducibility of your results, we recommend that if applicable you deposit your laboratory protocols in protocols.io, where a protocol can be assigned its own identifier (DOI) such that it can be cited independently in the future. For instructions see: http://journals.plos.org/plosone/s/submission-guidelines#loc-laboratory-protocols

We look forward to receiving your revised manuscript.

Kind regards,

Ming-Lung Yu, MD, PhD

Academic Editor

PLOS ONE

Journal Requirements:

2. Please include as Supporting Information, the full electronic search strategy and search terms for at least one database such that it could be repeated

Reviewers' comments:

Reviewer's Responses to Questions

**Comments to the Author**

1. Is the manuscript technically sound, and do the data support the conclusions?

Reviewer #1: Yes

Reviewer #2: Partly

2. Has the statistical analysis been performed appropriately and rigorously? 

Reviewer #1: Yes

Reviewer #2: Yes

3. Have the authors made all data underlying the findings in their manuscript fully available?

Reviewer #1: Yes

Reviewer #2: Yes

4. Is the manuscript presented in an intelligible fashion and written in standard English?

Reviewer #1: Yes

Reviewer #2: No

5. Review Comments to the Author

Reviewer #1: Authors estimated the prevalence rate of HCV infection among hemodialysis patients in Pakistan through systematic review and meta-analysis process. Comments are as followed.

1. The most favored disease prevalence of any kind derives from a national registry, and the same principle applies to the prevalence of HCV infection in hemodialysis patients. This study may have presented the second best way to estimate such a prevalence rate since in Pakistan there may be no national registry and the standard of medical practice was variable in different regions. Please clearly address such a situation in the Introduction and/or Discussion sections of this manuscript.

2. Authors have stated that there is significant heterogeneity in this study, and I understand that authors have tried to manage them with random-effect model. However, some cannot be statistically managed, for example, the definition of HCV infection -- 12 by ELIZA、4 by RNA、2 by CILA、and 2 unreported (total 20, not 19?). We know that about a quarter of anti-HCV (+) patients are actually HCV virus free in the blood. Such a fact will add another level of heterogeneity in your analysis. Please consider to discuss this further in your Discussion section.

3. Authors have tried to explain the high prevalence rate as “This is maybe due to the lack of education and awareness of HCV transmission, lack of scientifically and medically qualified, trained workers, lack of proper health infrastructure (use unsterilized instruments), etc.” However, in a more accepted way, it would additionally be (i) lack of strict infection control measure in the unit, (ii) inadequate use of erythropoietin, and (iii) inadequate screening of HCV/HIV for donated blood. Please address these more important issues in the text.

4. I do not quite agree with “The prevalence rate of HCV infection in hemodialysis patients is increasing with alarming rate…”, and the references quoted are out of date. Can authors update the information?

5. In page 4, what is “Siplimentry-2”? Is it “Supplementary 2”?

Reviewer #2: This article entitled of “The prevalence of hepatitis C virus in hemodialysis patients in Pakistan: a systematic review and meta-analysis” aimed to assess the pooled prevalence of HCV in hemodialysis patients in Pakistan." Several basic pitfalls render our reservation for publishing this article.

1. Information of the Figure 1 is not correctly display. The lines and the numbers need to be corrected.

2. The references number in the Table 1 and Appendix 2 is no match with the reference list.

3. Further English editing is suggested.

6. PLOS authors have the option to publish the peer review history of their article (what does this mean?). If published, this will include your full peer review and any attached files.

Reviewer #1: Yes: Jer-Ming Chang

Reviewer #2: No

---

## [Author Response · Author response to Decision Letter 0]

9 Mar 2020

Dear Editor,

Thanks a lot for your response and comments. All the comments of the reviews have been carefully considered and incorporated in the revised version. The point by point response is given below here and mentioned in the revised version in track changes, as given below.

Editor Comments:

In addition to the comments raised by reviewers, please discuss the prevalence of HCV infection among HD patients between Pakistan and the other countries in Asian-Pacific regions, especially Taiwan where incidence and prevalence of uremia ranking in the top 3 countries in the world.

Answer: The References (Taiwan and India) has been added in the Discussion section. 

Reviewer #1: Authors estimated the prevalence rate of HCV infection among hemodialysis patients in Pakistan through systematic review and meta-analysis process. Comments are as followed.

1. The most favored disease prevalence of any kind derives from a national registry, and the same principle applies to the prevalence of HCV infection in hemodialysis patients. This study may have presented the second best way to estimate such a prevalence rate since in Pakistan there may be no national registry and the standard of medical practice was variable in different regions. Please clearly address such a situation in the Introduction and/or Discussion sections of this manuscript.

Answer: Added accordingly in introduction and discussion section.

2. Authors have stated that there is significant heterogeneity in this study, and I understand that authors have tried to manage them with random-effect model. However, some cannot be statistically managed, for example, the definition of HCV infection -- 12 by ELIZA、4 by RNA、2 by CILA、and 2 unreported (total 20, not 19?). We know that about a quarter of anti-HCV (+) patients are actually HCV virus free in the blood. Such a fact will add another level of heterogeneity in your analysis. Please consider to discuss this further in your Discussion section.

Answer: Thank for your comments. The numbers of diagnostic tests are corrected accordingly. Further, the heterogeneity is further explored by using different diagnostic tests. Table 2 is extended accordingly and discussed in results section and discussion section.

3. Authors have tried to explain the high prevalence rate as “This is maybe due to the lack of education and awareness of HCV transmission, lack of scientifically and medically qualified, trained workers, lack of proper health infrastructure (use unsterilized instruments), etc.” However, in a more accepted way, it would additionally be (i) lack of strict infection control measure in the unit, (ii) inadequate use of erythropoietin, and (iii) inadequate screening of HCV/HIV for donated blood. Please address these more important issues in the text.

Answer: Thank you for your comment. Added, accordingly in discussion section.

4. I do not quite agree with “The prevalence rate of HCV infection in hemodialysis patients is increasing with alarming rate…”, and the references quoted are out of date. Can authors update the information?

Answer: The information is updated accordingly with some recent citation in introduction section. 

5. In page 4, what is “Siplimentry-2”? Is it “Supplementary 2”?

Answer: Thank you for correction. Corrected accordingly.

Reviewer #2: This article entitled of “The prevalence of hepatitis C virus in hemodialysis patients in Pakistan: a systematic review and meta-analysis” aimed to assess the pooled prevalence of HCV in hemodialysis patients in Pakistan." Several basic pitfalls render our reservation for publishing this article.

1. Information of the Figure 1 is not correctly display. The lines and the numbers need to be corrected.

Answer: Corrected accordingly.

2. The references number in the Table 1 and Appendix 2 is no match with the reference list.

Answer: Corrected accordingly. 

3. Further English editing is suggested.

Answer: English is edited significantly throughout the paper.

---

## [Decision Letter · Decision Letter 1]

10 Apr 2020

PONE-D-20-00973R1

The prevalence of hepatitis C virus in hemodialysis patients in Pakistan: a systematic review and meta-analysis

PLOS ONE

Dear Dr. Akhtar,

Thank you for submitting your manuscript to PLOS ONE. After careful consideration, we feel that it has merit but does not fully meet PLOS ONE’s publication criteria as it currently stands. Therefore, we invite you to submit a revised version of the manuscript that addresses the points raised during the review process.

Please revised the Figure 1.

We would appreciate receiving your revised manuscript by May 25 2020 11:59PM. To enhance the reproducibility of your results, we recommend that if applicable you deposit your laboratory protocols in protocols.io, where a protocol can be assigned its own identifier (DOI) such that it can be cited independently in the future. For instructions see: http://journals.plos.org/plosone/s/submission-guidelines#loc-laboratory-protocols

We look forward to receiving your revised manuscript.

Kind regards,

Ming-Lung Yu, MD, PhD

Academic Editor

PLOS ONE

Journal Requirements:

Additional Editor Comments (if provided):

Reviewers' comments:

Reviewer's Responses to Questions

**Comments to the Author**

1. If the authors have adequately addressed your comments raised in a previous round of review and you feel that this manuscript is now acceptable for publication, you may indicate that here to bypass the “Comments to the Author” section, enter your conflict of interest statement in the “Confidential to Editor” section, and submit your "Accept" recommendation.

Reviewer #1: All comments have been addressed

Reviewer #2: (No Response)

2. Is the manuscript technically sound, and do the data support the conclusions?

Reviewer #1: Yes

Reviewer #2: Yes

3. Has the statistical analysis been performed appropriately and rigorously? 

Reviewer #1: Yes

Reviewer #2: Yes

4. Have the authors made all data underlying the findings in their manuscript fully available?

Reviewer #1: Yes

Reviewer #2: Yes

5. Is the manuscript presented in an intelligible fashion and written in standard English?

Reviewer #1: Yes

Reviewer #2: Yes

6. Review Comments to the Author

Reviewer #1: (No Response)

Reviewer #2: The flowchart of the Figure 1 is not completely revised. The displaying number in the flowchart is misleading at the present status.

7. PLOS authors have the option to publish the peer review history of their article (what does this mean?). If published, this will include your full peer review and any attached files.

Reviewer #1: Yes: Jer-Ming Chang

Reviewer #2: No

---

## [Author Response · Author response to Decision Letter 1]

10 Apr 2020

Reviewer #2: 

The flowchart of the Figure 1 is not completely revised. The displaying number in the flowchart is misleading at the present status.

Answer: Thank you once again for your comment. The flowchart of the figure 1 has been revised accordingly.

---

## [Decision Letter · Decision Letter 2]

27 Apr 2020

The prevalence of hepatitis C virus in hemodialysis patients in Pakistan: a systematic review and meta-analysis

PONE-D-20-00973R2

Dear Dr. Akhtar,

We are pleased to inform you that your manuscript has been judged scientifically suitable for publication and will be formally accepted for publication once it complies with all outstanding technical requirements.

With kind regards,

Ming-Lung Yu, MD, PhD

Academic Editor

PLOS ONE

Additional Editor Comments (optional):

Reviewers' comments:

Reviewer's Responses to Questions

**Comments to the Author**

1. If the authors have adequately addressed your comments raised in a previous round of review and you feel that this manuscript is now acceptable for publication, you may indicate that here to bypass the “Comments to the Author” section, enter your conflict of interest statement in the “Confidential to Editor” section, and submit your "Accept" recommendation.

Reviewer #2: All comments have been addressed

2. Is the manuscript technically sound, and do the data support the conclusions?

Reviewer #2: (No Response)

3. Has the statistical analysis been performed appropriately and rigorously? 

Reviewer #2: (No Response)

4. Have the authors made all data underlying the findings in their manuscript fully available?

Reviewer #2: (No Response)

5. Is the manuscript presented in an intelligible fashion and written in standard English?

Reviewer #2: (No Response)

6. Review Comments to the Author

Reviewer #2: (No Response)

7. PLOS authors have the option to publish the peer review history of their article (what does this mean?). If published, this will include your full peer review and any attached files.

Reviewer #2: No

---

## [Editor Report · Acceptance letter]

5 May 2020

PONE-D-20-00973R2 

The prevalence of hepatitis C virus in hemodialysis patients in Pakistan: a systematic review and meta-analysis 

Dear Dr. Akhtar:

I am pleased to inform you that your manuscript has been deemed suitable for publication in PLOS ONE. Congratulations! Your manuscript is now with our production department. 

With kind regards,

on behalf of

Dr. Ming-Lung Yu 

Academic Editor

PLOS ONE